# Primary Tumor Location Is a Prognostic Factor for Intrahepatic Progression-Free Survival in Patients with Colorectal Liver Metastases Undergoing Portal Vein Embolization as Preparation for Major Hepatic Surgery

**DOI:** 10.3390/cancers12061638

**Published:** 2020-06-20

**Authors:** Lea Hitpass, Daniel Heise, Maximilian Schulze-Hagen, Federico Pedersoli, Florian Ulmer, Iakovos Amygdalos, Peter Isfort, Ulf Neumann, Christiane Kuhl, Philipp Bruners, Markus Zimmermann

**Affiliations:** 1Department of Diagnostic and Interventional Radiology, RWTH Aachen University Hospital, Pauwelsstrasse 30, D-52074 Aachen, Germany; lhitpass@ukaachen.de (L.H.); mschulze@ukaachen.de (M.S.-H.); fpedersoli@ukaachen.de (F.P.); isfort@ukaachen.de (P.I.); ckuhl@ukaachen.de (C.K.); pbruners@ukaachen.de (P.B.); 2Department of General, Visceral and Transplant Surgery, RWTH Aachen University Hospital, Pauwelsstrasse 30, D-52074 Aachen, Germany; dheise@ukaachen.de (D.H.); fulmer@ukaachen.de (F.U.); iamygdalos@ukaachen.de (I.A.); uneumann@ukaachen.de (U.N.)

**Keywords:** colorectal cancer, liver metastases, portal vein embolization, survival

## Abstract

The aim of this study was to identify prognostic factors affecting intrahepatic progression-free survival (ihPFS) and overall survival (OS) in patients with colorectal cancer liver metastases (CRCLM) undergoing portal vein embolization (PVE) and subsequent (extended) right hemihepatectomy. A total of 59 patients (mean age: 60.8 ± 9.3 years) with CRCLM who underwent PVE in preparation for right hemihepatectomy were included. IhPFS and OS after PVE were calculated using the Kaplan–Meier method. Cox regression analyses were conducted to investigate the association between the following factors and survival: patient age, laterality of the colorectal cancer (right- versus left-sided), tumor location (colon versus rectal cancer), time of occurrence of hepatic metastases (synchronous versus metachronous), baseline number and size of hepatic metastases, presence or absence of metastases in the future liver remnant (FLR) before PVE, preoperative carcinoembryogenic antigen (CEA) levels, time between PVE and surgery, history of neoadjuvant or adjuvant chemotherapy, and the presence or absence of extrahepatic disease before PVE. Median follow up was 18 months. The median ihPFS was 8.2 months (95% confidence interval: 6.2–10.2 months), and median OS was 34.1 months (95% confidence interval: 27.3–40.9 months). Laterality of the primary colorectal cancer was the only statistically significant predictor of ihPFS after PVE (hazard ratio (HR) = 2.242; 95% confidence interval: 1.125, 4.465; *p* = 0.022), with patients with right-sided colorectal cancer having significantly shorter median ihPFS than patients with left-sided cancer (4.0 ± 1.9 months versus 10.2 ± 1.5 months; log rank test: *p* = 0.018). Other factors, in particular also the presence or absence of additional metastases in the FLR, were not associated with intrahepatic progression-free survival. The presence of extrahepatic disease was associated with worse OS (HR = 3.050, 95% confidence interval: 1.247, 7.459; *p* = 0.015).

## 1. Introduction

Liver metastases in patients with colorectal cancer (CRC) are common and occur in about one-fourth of patients within the first 5 years after diagnosis of the primary tumor [1,2]. Surgery is the standard treatment in patients with resectable, oligometastatic liver disease, since complete resection prolongs overall survival and is potentially curative in 20–50% of patients [3,4,5].

Portal vein embolization (PVE) is routinely used to induce growth of the future liver remnant (FLR) in patients requiring extensive liver surgery, in order to prevent postoperative hepatic failure due to small-for-size syndrome. PVE has been shown to be an effective tool that allows patients who, without PVE, would not be considered candidates for surgical resection of liver metastases, to eventually be amenable to this possibly life-saving procedure [6,7]. 

However, previous studies have shown that PVE can stimulate post-interventional tumor growth in both liver lobes, especially in patients with a slow response to previous induction chemotherapy [8,9,10]. In 6–33% of patients, tumor progression in the FLR after PVE may preclude a curative resection and therefore negatively impact the overall outcome [11]. 

Moreover, there is significant variability of disease-free and overall survival among patients undergoing PVE and subsequent hepatectomy [12]. Accordingly, since PVE, as well as hepatectomy, are challenging procedures, there is a substantial clinical need to identify predictors that would help select patients who will benefit from such an intervention.

Therefore, the aim of this study was to identify factors that are associated with intrahepatic progression-free and overall survival of patients undergoing PVE and hepatectomy for colorectal cancer liver metastases (CRCLM).

## 2. Results

Out of a total of 162 patients who underwent PVE at our institution between June 2011 and December 2018, 59 met the inclusion criteria of this study (Figure 1). The mean follow up was 18 months (range: 4.7–95.1 months). 

### 2.1. Patient Demographics 

A summary of patient demographics is reported in Table 1.

#### 2.1.1. Tumor Mutation Status

*KRAS* mutation status was available in 41/59 patients (70%) and was unknown in the remaining 18 patients (30%). There were 12 patients with a *KRAS* mutation (12/41, 29%). In the group of patients with right-sided CRC (*n* = 14), 6 patients (43%) had a *KRAS* mutation, 4 patients (29%) had a wild-type *KRAS* gene, and the *KRAS* mutation status was unknown in the remaining 4 patients. In the group of patients with left-sided CRC (*n* = 45), 6 patients (13%) had a *KRAS* mutation, 25 patients (61%) had a wild-type *KRAS* gene, and mutation status was unknown in the remaining 14 patients.

#### 2.1.2. Treatment of the Primary Tumor

In the majority of patients (47/59, 80%), resection of the primary tumor was performed before PVE and subsequent hepatic resection. In the remaining 12 patients (20%), a “liver-first approach” was pursued, and these patients were scheduled to undergo bowel surgery after PVE and (extended) right hemihepatectomy. Due to systemic disease progression, resection of the primary tumor was cancelled in 4 out of these 12 patients. Of the 19 patients with rectal cancer, 7 underwent short-term neoadjuvant chemoradiation before resection of the primary tumor, 6 received neoadjuvant chemotherapy, and the remaining 6 did not receive any neoadjuvant treatment before resection of the primary tumor.

#### 2.1.3. Extrahepatic Disease before PVE

Eleven out of 59 patients (19%) had limited extrahepatic disease before undergoing PVE in preparation for subsequent hepatic resection. Five patients had isolated retroperitoneal or celiac lymph node metastases, which were surgically removed during liver resection and histopathologically confirmed. Two patients had a solitary pulmonary metastasis and were scheduled to undergo resection of these pulmonary metastases after resection of the liver metastases. Three patients showed small pulmonary nodules (<5 mm) on the baseline staging computed tomography (CT), which were initially classified as non-specific but turned out to be metastases over the course of the post-surgical follow up. Finally, one patient had a metastasis in the left adrenal gland, which was resected and histopathologically confirmed.

#### 2.1.4. Treatment with Chemotherapy before and after PVE and Liver Surgery

Thirty-six patients (61%) had received neoadjuvant treatment with chemotherapy within 6 months before PVE and right hemihepatectomy. Eleven patients (19%) received 3–6 cycles of chemotherapy between PVE and surgery with the intention to downstage and improve resectability. Thirteen patients (22%) received adjuvant chemotherapy after right hemihepatectomy. Chemotherapy regimens were chosen in close adherence to current international practice and included FOLFOX, FOLFIRI, FOLFIRINOX, CapeOx, Capecitabine, Bevacizumab, Cetuximab, and Panitumumab.

### 2.2. Follow up after PVE

After PVE, 8 patients out of 59 patients (14%) developed new liver metastases in the FLR before the planned date of surgery. In 7 of these, hepatectomy was canceled because the local tumor burden in the FLR was considered prohibitive. In one patient, two newly occurred metastasis in the FLR after PVE were successfully treated by local ablation, and surgery was completed as planned. In 3 additional patients (5%), extrahepatic disease was identified during surgery, and the planned hepatic resection was aborted. Accordingly, 49 patients underwent right hemihepatectomy or trisectorectomy after a mean of 49 ± 44 days after PVE.

In the follow up after surgery, 27 patients (27/49; 55%) developed intrahepatic tumor progression in the liver remnant after a mean of 7.6 ± 4.9 months (range: 1–18 months).

The median intrahepatic progression-free survival after PVE among all 59 patients—regardless of whether they proceeded to surgery or not—was 8.2 months (95% confidence interval: 6.2–10.2 months). 

### 2.3. Association of Clinical Features with Intrahepatic Progression-Free Survival

Univariable Cox regression analyses revealed that the laterality of the primary CRC was the only statistically significant predictor of intrahepatic progression-free survival, with a hazard ratio of 2.242 (95% confidence interval: 1.125, 4.465; *p* = 0.022) (Table 2).

Patients with a right-sided primary tumor had a median intrahepatic progression-free survival (ihPFS) of 4.0 ± 1.9 months, which was significantly shorter than the median ihPFS of 10.2 ± 1.5 months for patients with a left-sided primary (log-rank test: *p* = 0.018) (Figure 2). 

There was no statistically significant difference in median ihPFS for patients who had liver metastases in the FLR before PVE versus those who did not (7.4 ± 1.8 months versus 10.2 ± 1.3 months, log-rank test: *p* = 0.265) (Figure 3).

### 2.4. Association of Clinical Features with Overall Survival

The Kaplan–Meier estimate for overall survival after PVE among all 59 patients was 34.1 months (95% confidence interval: 27.3–40.9 months). The 1-year, 3-year, and 5-year overall survival rates were 79%, 41%, and 24%, respectively. There was no statistically significant difference in overall survival between patients with right-sided CRC and those with a left-sided primary tumor (23.5 ± 11.6 versus 34.1 ± 3.5 months, log-rank test: *p* = 0.942). Employing univariable Cox regression analyses, the presence of extrahepatic disease was statistically significantly associated with a worse overall patient survival (hazard ratio = 3.050, 95% confidence interval: 1.247, 7.459; *p* = 0.015). Patients with extrahepatic disease had a median overall survival (OS) of 28.3 ± 11.1 months, which was significantly shorter than the median OS of 35.2 ± 3.9 months for patients without evidence of extrahepatic disease on the pre-interventional staging CT (log-rank test: *p* = 0.01) (Figure 4 and Table 3).

## 3. Discussion

The oncological outcome of patients with CRCLM undergoing PVE in preparation for subsequent (extended) right hemihepatectomy is highly variable, with some patients developing new hepatic metastases in the FLR as early as a few weeks after PVE, even before hepatic resection, while others remain disease-free for several years after surgery. This study investigated factors that are associated with ihPFS in these patients, in order to identify risk factors associated with poor clinical outcome. The results of this small study on a limited number of patients demonstrate that ihPFS is mainly driven by the location of the primary tumor, in other words: by biology of the primary tumor, rather than by the local intrahepatic tumor burden, or pre-existing tumor in the FLR. Patients with right-sided CRCs had an approximately 6 months shorter ihPFS after PVE than patients with a left-sided primary tumor. On the other hand, none of the remaining factors investigated such as for example size and number of liver metastases before PVE, pre-existing tumor in the future liver remnant (FLR) or the time of detection liver metastases (synchronous or metachronous) and pre- or postoperative systemic chemotherapy treatment were associated with ihPFS—which means that these factors either do not, or at least not to the same extent, impact ihPFS as does tumor biology. 

The results of this study in terms of ihPFS are in good agreement with the observation that right-sided and left-sided CRCs differ with regard to their respective molecular and biological features, with right-sided CRCs being biologically more aggressive cancers, occurring in younger patients, and being associated with a reduced overall survival compared to their left-sided counterparts [13,14]. The exact underlying molecular mechanism for this difference in aggressiveness remain unclear, although we found that that *KRAS* mutations occurred more often in patients with right-sided CRCs than in patients with left-sided CRCs in our cohort—a finding that is in line with previously published studies [15,16]. Therefore, one possible explanation could be that tumors with certain molecular features, such as for example a mutation in the *KRAS* oncogene, produce more micrometastases in the liver, which are preoperatively undetectable by current imaging methods due to their small size but become visible over time with tumor growth and are therefore responsible for rapid tumor “recurrence” after surgery. 

In our small cohort, Kaplan–Meier estimates for overall survival did also differ for patients with right-sided versus left-sided primary cancers (23.5 ± 11.6 versus 34.1 ± 3.5 months), but this did not reach statistical significance. This is likely due to the fact that this study was underpowered for this sub-analysis. It is particularly encouraging that the presence of metastases in the FLR before PVE had no impact on progression-free or overall survival, which supports aggressive treatment strategies with clearing of the FLR by percutaneous ablation or atypical resection for patients with bilobar CRCLM. However, the presence of even limited extrahepatic disease such as isolated abdominal lymph node metastases or solitary lung metastases was expectably associated with a worse overall survival compared to patients without such extrahepatic disease (28.3 ± 11.1 versus 35.2 ± 3.9 months), which is in line with results from previous studies [17,18,19].

We hypothesized that the size or number of hepatic metastases could be associated with ihPFS since more and larger metastases should be either a result of a higher biological aggressiveness of the primary tumor, or a result of delayed diagnosis. In both cases, the risk of undetectable micrometastases in the FLR should—at least in theory—be higher, which could lead to earlier tumor “recurrence” after PVE and surgery. However, the results of the Cox regression analyses in this patient cohort do not support this theory, since baseline size and number of hepatic metastases were not associated with intrahepatic progression-free or overall survival.

If, after PVE, a patient recurs in the FLR or outside the liver; then usually, the planned surgical treatment is abandoned. Previous studies demonstrated that up to 25% of patients develop such progressive disease in the FLR after PVE [20]. The relatively small number of patients who were unable to undergo resection due to disease progression in the FLR in the current study (7/59, 12%) compares favorably with these prior observations. 

However, the median intrahepatic progression-free survival of 8.2 months and the median overall survival of 34.1 months observed in this study were inferior to survival rates previously published in a meta-analysis, where an average disease-free survival of 15.2 months and overall survival of 38.9 months were reported. [20]. We believe that the difference may be explainable by the fact that due to the emerging evidence on the utility of aggressive local treatment in patients with CRCLM, patients with a much higher local tumor load are accepted as surgical candidates today than at the time when the data for the meta-analysis had been collected. 

Despite the fact that a significant amount of patients experienced tumor recurrence at some point after PVE and liver resection, we think the 5-year overall survival rate of 24% is nevertheless encouraging—especially when considering the high initial tumor burden in this patient cohort. The results of our study suggest that even in the setting of extensive hepatic disease, long-term (disease-free) survival can be achieved in some patients and a curative treatment approach should therefore be pursued whenever possible. 

Limitations of this study include its retrospective design and the small size of the patient cohort which makes subgroup analyses susceptible to random statistical errors. Accordingly, further studies are needed to confirm our results, and verification of these results in a larger patient cohort is warranted. Additionally, *KRAS*, *NRAS*, and *BRAF* mutation status as well as microsatellite instability status, all of which may significantly impact the course of disease, was not available in all patients of this cohort, and could therefore not be analyzed regarding their association with survival. Finally, overall survival times are likely biased by other treatments received by patients with disease recurrence, such as re-resection, local ablation, trans-arterial chemoembolization, radioembolization, and systemic chemotherapy. 

## 4. Materials and Methods

Approval for this retrospective study was granted by the institutional review board (IRB, internal reference no. EK 152/19). Written informed consent for all interventional procedures was obtained from all patients.

### 4.1. Description of Patient Cohort

All patients with CRCLM who underwent PVE in preparation for subsequent right hemihepatectomy or extended right hemihepatectomy (right trisectionectomy) between June 2011 and December 2018 at our institution were identified from the institutional database. Exclusion criteria were the absence of a contrast-enhanced CT or magnetic resonance imaging (MRI) scan within 3 weeks prior to PVE, death due to postoperative complications within 30 days after surgery, and follow up of less than three months after surgery (for patients who eventually proceeded to surgery) or after PVE (for patients who did not).

### 4.2. Portal Vein Embolization Procedure

All oncological treatment decisions at our institution are made by consensus in a multidisciplinary tumor board attended by hepatobiliary surgeons, oncologists, radiotherapists, pathologists, radiologists, and interventional radiologists. Only patients with liver-dominant, resectable disease and a future liver remnant of ≤20% of the total functional liver volume (≤40% in patients with a history of chemotherapy) are candidates for preoperative portal vein embolization. In practice, this means that mostly patients with a planned extended right hemihepatectomy/right trisectionectomy or patients with a history of chemotherapy in whom a right hemihepatectomy is planned receive a PVE at our institution.

The standardized interventional protocol pursued an ipsilateral approach. Under ultrasound guidance, a right portal vein radicle was punctured, and a 6F sheath (Brite tip, Cordis, Hialeah, FL, USA) was placed into the main stem of the portal vein. Then, a 5F-SOS or Sidewinder catheter (Cook Medical, Bloomington, IN, USA) was introduced and placed with the tip into the main stem of the right portal vein. Then, all branches of the right portal vein (segments V–VIII) were embolized in succession, using a 4:1 mixture of n-butyl-2-cyanoacrylate (Histoacryl, Braun, Melsungen, Germany) and iodized oil (Lipiodol, Guerbet, Villepinte, France) through a 2.7F-microcatheter (Renegade, Boston Scientific, Marlborough, MA, USA). Regardless of whether a resection of segment IV is planned (extended right hemihepatectomy) or not (right hemihepatectomy), the portal vein branch of segment IV is routinely spared from embolization at our institution, since this gives the surgeon more flexibility regarding the extent of the resection of segment IV intraoperatively. Additionally, the avoidance of embolization of segment IV branches reduces the risk of non-target embolization of the left main portal vein and segment II/III branches, whilst the need for the embolization of segment IV appears questionable; at least one previous study has shown that the hypertrophy of segments II and III is similar after a right-sided PVE versus a right-sided PVE plus the embolization of segment IV [21]. After successful embolization of all right hepatic portal vein branches, the puncture tract was occluded through embolization with the aforementioned mixture during simultaneous removal of the sheath, macrocatheter, and microcatheter.

### 4.3. Pre-Interventional Imaging

The standard protocol included a contrast-enhanced whole-body CT and either a multiphasic CT examination of the liver or a liver MRI with liver-specific contrast agent (gadoxetic acid) no more than 3 weeks before the planned PVE to assess extrahepatic metastases, to define surgical resectability, and plan the PVE as well as additional local treatments of metastases in the future liver remnant by means of either local ablation or minor atypical surgical resection.

### 4.4. Post-Interventional Follow up

Two to four weeks after PVE, patients received a contrast-enhanced liver CT or liver MRI to assess hypertrophy of the future liver remnant, and to search for new metastases in the FLR or at extrahepatic sites. 

Before eventually proceeding with surgery, all surgical candidates would undergo whole-body re-staging, if the most recent staging examination was older than 4 weeks. Re-staging of the liver was performed no more than 2 weeks before surgery. Patients who showed unresectable progressive disease according to Response Evaluation Criteria In Solid Tumors (RECIST) criteria on these re-staging examinations were no longer considered surgical candidates and would pursue alternative treatment options (usually chemotherapy). 

### 4.5. Post-Surgical Follow up

After (extended) right hemihepatectomy, adjuvant treatment with chemotherapy was given based on the individual recommendation of a multidisciplinary tumor board. Patients were followed according to a standardized protocol consisting of clinical examination, laboratory tests (liver function and tumor markers), and contrast-enhanced imaging (CT or MRI) every 3 months. 

When disease recurrence occurred, patients were treated with re-resection, local ablation, transarterial chemotherapy, transarterial radioembolization, or palliative chemotherapy, depending on individual circumstances such as the extent and location of disease recurrence.

### 4.6. Data Collection and Analysis

Baseline patient and treatment characteristics were retrospectively collected from electronic medical records and each patient´s baseline (pre-interventional) imaging examinations were reviewed. The following parameters were selected for further analysis:(a)patient age(b)laterality of the primary tumor (right-sided colon cancer, arising from caecum, ascending colon and transverse colon versus left-sided colon cancer, arising from descending colon, sigmoid colon, and rectum) [22](c)location of the primary tumor (colon cancer versus rectal cancer)(d)time interval between diagnosis of the primary colorectal cancer and detection of hepatic metastases (synchronous versus metachronous metastases)(e)largest axial diameter of the hepatic metastases on baseline imaging(f)number of hepatic metastases on baseline imaging(g)presence or absence of hepatic metastases in the FLR before PVE (these were subsequently either treated by local ablation or surgically resected by means of atypical resection before or during right hemihepatectomy)(h)preoperative serum levels of carcinoembryogenic antigen (CEA)(i)time (number of days) between PVE and right hemihepatectomy(j)history of chemotherapy treatment within 6 months before and after hepatic resection(k)presence or absence of (limited) extrahepatic disease on the baseline imaging before PVE

### 4.7. Statistics

The Kaplan–Meier method was used to calculate intrahepatic progression-free and overall survival after PVE, and results were reported as medians, including their 95% confidence intervals. Survival estimates were compared using the log-rank test wherever applicable.

Univariable Cox regression analyses were performed to identify possible independent predictors for intrahepatic progression-free and overall survival among the abovementioned variables. Continuous variables were summarized using proportions, means, and medians. All statistical analyses were performed using Excel (Microsoft Office 2016, Redmond, WA, USA) and SPSS (Version 25, IBM, Armonk, NY, USA).

## 5. Conclusions

In conclusion, we found that the recurrence of CRCLMs in the liver remnant after PVE and (extended) right hemihepatectomy mainly depends on the biology of the primary CRC, rather than on local tumor load. Intrahepatic recurrence occurs significantly more often and earlier in patients with hepatic metastases of a right-sided CRC compared to patients with a left-sided CRC. Additionally, the presence of even limited extrahepatic metastases was associated with worse overall survival. However, further studies are necessary to confirm our findings in larger patient cohorts and to investigate potential underlying mechanisms responsible for tumor aggressiveness, such as certain molecular or genetic tumor features.

## Figures and Tables

**Figure 1 cancers-12-01638-f001:**
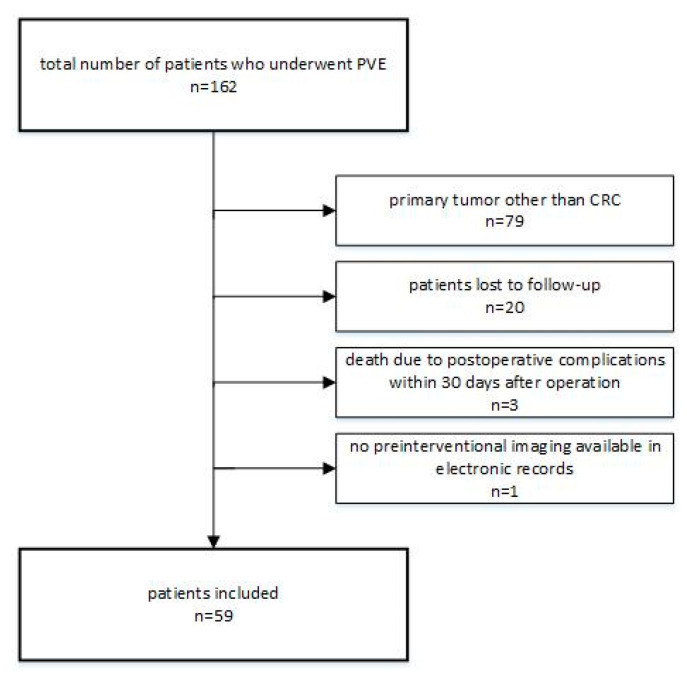
Flow chart of study inclusion/exclusion.

**Figure 2 cancers-12-01638-f002:**
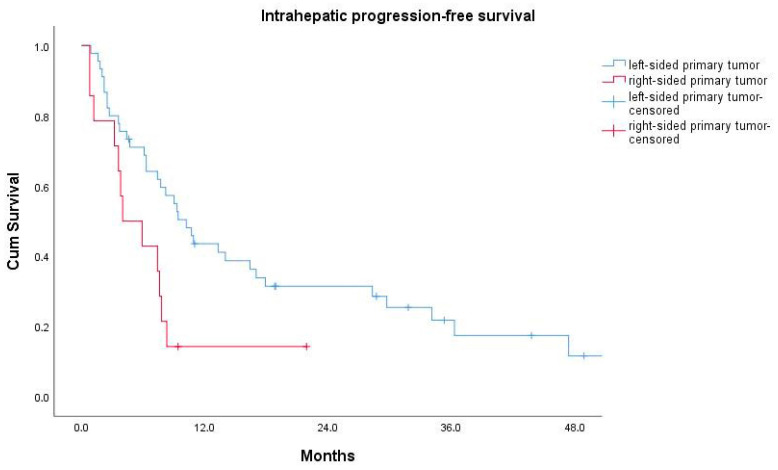
Kaplan–Meier curves for intrahepatic progression-free survival for patients with a left-sided (blue curve) vs. those with a right-sided (red curve) primary tumor.

**Figure 3 cancers-12-01638-f003:**
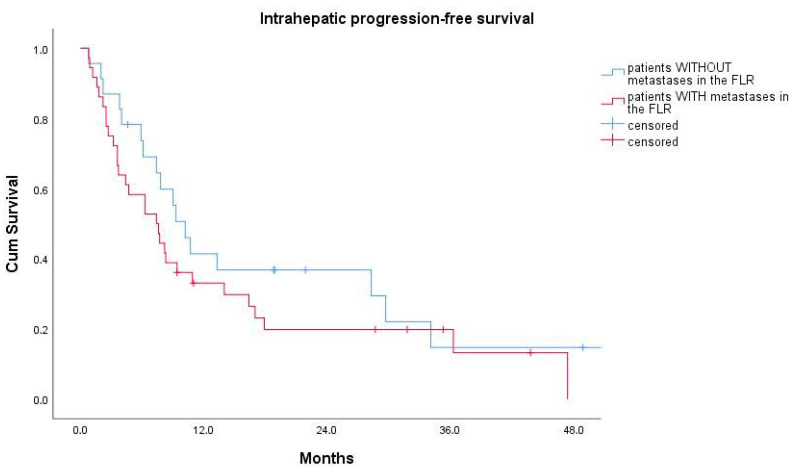
Kaplan–Meier curves for intrahepatic progression-free survival for patients with (red curve) and without (blue curve) liver metastases in the FLR before portal vein embolization.

**Figure 4 cancers-12-01638-f004:**
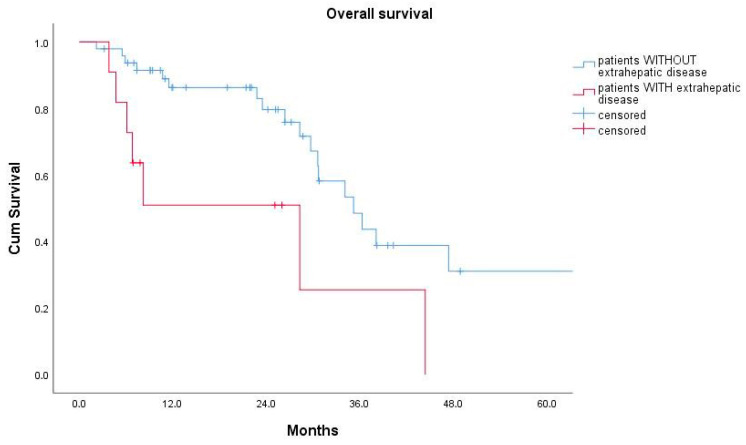
Kaplan–Meier curves for overall survival for patients without (blue) and with (red) evidence of limited extrahepatic disease on pre-interventional imaging.

**Table 1 cancers-12-01638-t001:** Baseline patient characteristics. CEA: carcinoembryogenic antigen, FLR: future liver remnant, PVE: portal vein embolization.

Total Number of Patients	*n* = 59
Age (years)	60.8 ± 9.3
Male/Female	43/16
Number of hepatic metastases	5.0 ± 3.0 (range: 1–13)
Size of largest hepatic metastasic lesion	4.0 ± 2.1 cm (range: 0.7–11.3 cm)
Preoperative serum CEA level	210.7 ± 812.1 ng/mL
Time between PVE and surgery	49 ± 44 days
Tumor laterality	
Right-sided colorectal cancer	14
Left-sided colorectal cancer	45
Primary tumor location	
Colon	40
Rectum	19
Timing of occurrence of hepatic metastases	
synchronous	53
metachronous	6
Presence of metastases in FLR	
Yes	34
No	25
Chemotherapy treatment **	
Within 6 months leading up to PVE	36
Between PVE and surgery	11
Within 6 months after surgery	13
Presence of extrahepatic disease	
Yes	11
No	48

* Mean ± standard deviation is reported for age, number of hepatic metastases, and size of the largest metastatic lesion. ** Some patients received chemotherapy at more than one point in time, while others did not receive chemotherapy at all.

**Table 2 cancers-12-01638-t002:** Results of the univariable Cox regression analysis for intrahepatic progression-free survival. CEA: carcinoembryogenic antigen, PVE: portal vein embolization.

Factor	Hazard Ratio (95% Confidence Interval)	*p* Value
Age	0.989 (0.959, 1.021)	0.654
Laterality of the primary colorectal cancer (right-sided vs. left-sided)	2.242 (1.125, 4.465)	0.022
Location of the primary tumor (colon vs. rectal cancer)	0.674 (0.363, 1.250)	0.211
Time interval between diagnosis of the primary colorectal cancer and detection of hepatic metastases (synchronous vs. metachronous)	3.114 (0.932, 10.409)	0.065
Size of the largest metastasis at baseline	1.021 (0.904, 1.153)	0.736
Number of metastases at baseline	0.998 (0.905, 1.102)	0.974
Presence or absence of metastases in the FLR	1.402 (0.771, 2.547)	0.268
Preoperative serum CEA level	1.000 (1.000,1.000)	0.736
Time between PVE and right hemihepatectomy	1.002 (0.995, 1.009)	0.534
Neoadjuvant chemotherapy prior to surgery	1.539 (0.797, 2.970)	0.199
Adjuvant chemotherapy after surgery	1.538 (0.740, 3.197)	0.248
Presence of extrahepatic disease	1.712 (0.860, 3.410)	0.126

**Table 3 cancers-12-01638-t003:** Results of the univariable Cox regression analysis for overall survival. CEA: carcinoembryogenic antigen, PVE: portal vein embolization.

Factor	Hazard Ratio (95% Confidence Interval)	*p* Value
Age	1.021 (0.969, 1.056)	0.591
Laterality of the primary colorectal cancer (right-sided vs. left-sided)	0.954 (0.269, 3.388)	0.942
Location of the primary tumor (colon vs. rectal cancer)	1.378 (0.621, 3.056)	0.430
Time interval between diagnosis of the primary colorectal cancer and detection of hepatic metastases (synchronous vs. metachronous)	4.712 (0.628, 35.331)	0.132
Size of the largest metastasis at baseline	1.138 (0.959, 1.351)	0.139
Number of metastases at baseline	0.998 (0.869, 1.145)	0.927
Presence or absence of metastases in the FLR	0.639 (0.290, 1.408)	0.267
Preoperative serum CEA levels	1.000 (0.999, 1.001)	0.972
Time between PVE and right hemihepatectomy	0.993 (0.977, 1.010)	0.427
Neoadjuvant chemotherapy treatment	0.813 (0.358, 1.847)	0.622
Adjuvant chemotherapy treatment	0.783 (0.290, 2.115)	0.630
Presence of extrahepatic disease on baseline imaging	3.050 (1.247, 7.459)	0.015

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
