# Peer review of "Primary Tumor Location Is a Prognostic Factor for Intrahepatic Progression-Free Survival in Patients with Colorectal Liver Metastases Undergoing Portal Vein Embolization as Preparation for Major Hepatic Surgery"

_cancers, 2020, doi:10.3390/cancers12061638_

Round 1

Reviewer 1 Report

The Paper “Primary Tumour Location is a Prognostic Factor for Intrahepatic Progression-Free Survival in Patients with Colorectal Liver Metastases Undergoing Portal Vein Embolization as Preparation for Major Hepatic Surgery” published by Hitpass et al. aims to identify prognostic factors anticipating the outcome of CRCLM Patients undergoing PVE and subsequent hemi-hepatectomy.
Hitpass et al. present the correlation between primary tumour location and intrahepatic progression free survival as their main finding.

Major Comments

  • The authors only refer to the site of the primary carcinoma as ‘left or right sided colon cancers’ (see table 1, below line 66) leaving the reader in question, if left sided carcinomas include the rectum or not.
  • The authors analysed the data of 59 patients, that underwent right hemi-hepatectomy (line 21) without explaining the reason why they focused on this cohort and did not include e.g. left hemi-hepatectomy.
  • Hitpass et al. are presenting a significant correlation between right sided primary colon cancer and a comparatively worse ihPFS (see table 2, below line 91, p-value 0.022) supporting the well-known fact, that right sided colon cancer is more aggressive than left sided colon cancer (line 122). However, an comprehensive reasoning for which molecular mechanisms or route of metastasis could be involved in this event is lacking when authors are concluding that the recurrence of CRCLM is based on biology (line 238).
  • Similarly, Hitpass et al. cite that PVE can stimulate tumour growth in the FLR (line 47), and point out that PVE and Hepatectomy are challenging procedures and predictive factors have to be found to avoid unnecessary surgery (line 53 to 54), however do not provide an approach, on what the molecular drivers behind tumour growth in FLR could be.

Minor Comments

line 21: “A total of 59 patients”
line 22: blank space missing before “years”
line 46: blank space missing before citation numbers
line 49: abbreviation FLR ought to be used
line 79: “an” should be deleted
line 84: no comma
line 109: “On the univariable Cox..” should be replaced by e.g. “Employing univariable Cox...”
line 117: abbreviation FLR ought to be used
line 119: abbreviation ihPFS ought to be used
line 123: abbreviation FLR ought to be used
Figure 1:  electronic
line 195: “BBraun”

Author Response

Response to comments of reviewer 1

Major Comments

  • The authors only refer to the site of the primary carcinoma as ‘left or right sided colon cancers’ (see table 1, below line 66) leaving the reader in question, if left sided carcinomas include the rectum or not.

Response:

We classified all colon cancers based on a previously published and well-established definition1: “Right-sided colon cancer” refers to all cancers originating proximal to the splenic flexure (caecum, ascending colon, transverse colon) and “left-sided colon cancer” refers to all tumors arising distally to this site (descending colon, sigmoid colon and rectum). We have added this definition in the Materials and Methods section (paragraph 4.6, lines 258-259).

1 Iacopetta B. Are there two sides to colorectal cancer? Int J Cancer. 2002; 101(5):403-8.

  • The authors analysed the data of 59 patients, that underwent right hemi-hepatectomy (line 21) without explaining the reason why they focused on this cohort and did not include e.g. left hemi-hepatectomy.

Response:

With the right liver lobe comprising approximately 75% of the total liver volume, right hemi-hepatectomy is a much larger and often more complicated surgical procedure compared to resection of the usually much smaller left liver lobe. In order to avoid post-operative liver insufficiency, induction of hypertrophy of the left liver lobe by means of PVE is often necessary before right hemi-hepatectomy, while hypertrophy-induction of the right liver lobe before left hemihepatectomy is rarely needed. So in summary, a curative treatment approach in patients with colorectal liver metastases requiring right-hemihepatectomy is significantly more time-consuming and complicated, hence we focused in our study on this specific patient cohort.

  • Hitpass et al. are presenting a significant correlation between right sided primary colon cancer and a comparatively worse ihPFS (see table 2, below line 91, p-value 0.022) supporting the well-known fact, that right sided colon cancer is more aggressive than left sided colon cancer (line 122). However, an comprehensive reasoning for which molecular mechanisms or route of metastasis could be involved in this event is lacking when authors are concluding that the recurrence of CRCLM is based on biology (line 238).

Response:

This is indeed a very interesting point, although our study probably does not allow us to draw any conclusions regarding the underlying molecular mechanisms which are responsible for the difference in clinical outcome among our patients. Anyhow, one hypothesis would be that certain tumors produce and spread more micrometastases throughout the body, which are undetectable by current imaging methods as long as they do not reach a certain size (e.g. metastases under 1-2mm will probably not be detected by MRI, for CT the threshold would be even higher, probably around 5mm). We would argue that right-sided colon cancers produce more of these micrometastases, maybe based on certain mutations such as of the KRAS oncogene, and that these micrometastases will be present but undetected in the FLR before surgery. With some time to grow after surgery, they will however become detectable by current clinical imaging modalities and therefore lead to early tumor “recurrence”. We have expanded the second paragraph in the discussion section and included this hypothesis (lines 153-160).

  • Similarly, Hitpass et al. cite that PVE can stimulate tumour growth in the FLR (line 47), and point out that PVE and Hepatectomy are challenging procedures and predictive factors have to be found to avoid unnecessary surgery (line 53 to 54), however do not provide an approach, on what the molecular drivers behind tumour growth in FLR could be.

Response:

In the case of stimulation of tumor growth in the FLR by PVE there is another factor involved: Hypertrophy of the FLR is induced by increased portal blood flow to the FLR, as the blood volume that would normally flow into the right side of the liver now flows to the left side (FLR) as well. Growth of the FLR is then induced by mediators such as TNF-alpha, Interleukin-6, hepatocyte growth factor (HGF), epidermal growth factor (EGF), fibroblast growth factor(FGF) and transforming growth factor(TGF).1 It seems logical that these mediators do not only stimulate growth of normal hepatocytes but also may stimulate growth of any (micro-)metastases present in the FLR. However, although we agree that this is a very interesting topic, it is also beyond the scope of the current study.

1 Le Roy B, Dupré A, Gallon A, Chabrot P, Gagnière J, Buc E. Liver hypertrophy: Underlying mechanisms and promoting procedures before major hepatectomy. J Visc Surg. 2018;155(5):393‐401. doi:10.1016/j.jviscsurg.2018.03.005

Minor Comments

line 21: “A total of 59 patients”

line 22: blank space missing before “years”

line 46: blank space missing before citation numbers (now line 47)

line 49: abbreviation FLR ought to be used (now line 50)

line 79: “an” should be deleted (now line 96)

line 84: no comma (now line 101)

line 109: “On the univariable Cox..” should be replaced by e.g. “Employing univariable Cox...” (now line 127)

line 117: abbreviation FLR ought to be used (now line 136)

line 119: abbreviation ihPFS ought to be used (now line 138)

line 123: abbreviation FLR ought to be used (now line 142)

Figure 1:  electronic

line 195: “BBraun” (now line 224)

Response:

All minor comments have been adressed and corrected in the manuscript.

Thank you very much for your time and effort to review our manuscript.

Reviewer 2 Report

Dear Authors,

I have read with interest this manuscript, which concerns an interesting topic in HPB surgery. In particular in surgery for colorectal liver metastases. The manuscript of overall well written and might be of interest to the readers of Cancers. However, there are some issues worthy to be notified and fixed before considering for publication:

  1. In the abstract the acronym CRCLM does not fit with the words colorectal liver metastases. The word cancer is missing.
  2. Again, hazard ratio may be abbreviated in HR. Otherwise please do not abbreviate CI. Authors should be consistent with style and abbreviations.
  3. The number of patients is low (59). More importantly, it is unclear if the number of patients really analysed is 59. Indeed, among 59 patients 7 developed new CLMs after PVE, and 3 had the finding of extrahepatic diseases during laparotomy. It seems that 10 patients were then excluded. Were 49 those patients included in the statistical analyses (survival analyses)? Or the authors did an intention-to-treat analysis by including 59 patients?
  4. The variable “time from PVE date to surgery date” should be included in the analysis. In fact, surgery was done after 49±44 days after PVE. Such large SD indicates that some patients had surgery many weeks after the PVE, while some others very soon.
  5. Please add preoperative tumor markers, at least CEA as a variable.
  6. Please clarify if all patients underwent surgery after restaging of the disease and if such disease was under control from the oncological standpoint. Please use standard radiological criteria to make the reader understand this important point.
  7. Please add, if available, KRAS NRAS and BRAF status. Otherwise please add this as a strong limit of the authors’ findings interpretation.
  8. Please add the types of chemotherapy performed.
  9. Which is the meaning of considering intrahepatic recurrence so important when the overall survival is not impacted by any of the variables considered?
  10. Please add the type of postoperative treatments. Were those patients who developed intrahepatic recurrence treated with more aggressive therapies? Otherwise, it seems that some others variables really impacted overall survival and their misidentification represents a strong bias – also for intrahepatic recurrence.
  11. It is unclear when bowel surgery was performed. Before? Before PVE but together with liver surgery? After liver surgery? This information is not secondary since the main authors’ finding regards the primary colic tumor.
  12. Was adjuvant chemotherapy that therapy performed after the bowel surgery or after the liver surgery?

Author Response

Response to comments of reviewer 2

In the abstract the acronym CRCLM does not fit with the words colorectal liver metastases. The word cancer is missing.

Response:

This was edited as suggested (line 20)

Again, hazard ratio may be abbreviated in HR. Otherwise please do not abbreviate CI. Authors should be consistent with style and abbreviations.

Response:

This was edited as suggested in the abstract (line 29 and 31) and the rest of the manuscript.

The number of patients is low (59). More importantly, it is unclear if the number of patients really analysed is 59. Indeed, among 59 patients 7 developed new CLMs after PVE, and 3 had the finding of extrahepatic diseases during laparotomy. It seems that 10 patients were then excluded. Were 49 those patients included in the statistical analyses (survival analyses)? Or the authors did an intention-to-treat analysis by including 59 patients?

Response:

We indeed performed an “intention-to-treat-analysis” and included all 59 patients in the survival analyses for intrahepatic progression-free survival (ihPFS) and overall survival (OS). For ihPFS, this is mentioned in lines 102-104 and for OS, we added this info in line 123.

The variable “time from PVE date to surgery date” should be included in the analysis. In fact, surgery was done after 49±44 days after PVE. Such large SD indicates that some patients had surgery many weeks after the PVE, while some others very soon.

Response:

Patients that develop sufficient hypertrophy of the FLR on the follow-CT after PVE (usually around 3 weeks post PVE) normally undergo resection about 1 month after PVE at our institution. However, in some patients, the growth rate of the FLR is slower and therefore hypertrophy of the FLR is insufficient on the initial follow-up CT. These patients often will receive an additional 3-6 cycles of chemotherapy to keep the disease under control while waiting for the FLR to reach a sufficiently large size. In this cohort, there are 11 patients of these patients, which subsequently were resected several months after PVE which leads to the large standard deviation for the variable “time from PVE date to surgery date”.

We have performed additional Cox Regression analyses with “time between PVE and surgery” as an independent variable as suggested, however we found not statistically significant association with ihPFS or OS. (Table 2+3 and line 269)

Please add preoperative tumor markers, at least CEA as a variable.

Response:

We have included preoperative serum CEA levels in the Cox regression analyses as suggested. CEA levels however do not seem to have any prognostic value with regard to ihPFS or OS. (Table 2+3 and line 268)

Please clarify if all patients underwent surgery after restaging of the disease and if such disease was under control from the oncological standpoint. Please use standard radiological criteria to make the reader understand this important point.

Response:

All patients in this study underwent major liver surgery with curative intent and for this purpose, it is of course necessary to exclude extrahepatic disease which would negate such a curative treatment approach. Therefore, all surgical candidates are required to have a recent whole-body staging before proceeding with surgery, with “recent” meaning no older than 4 weeks before surgery. So patients would undergo whole-body restaging before surgery, if the most recent staging was older than 4 weeks.

Re-staging of the liver is however always routinely performed even closer before surgery, since follow-up imaging after PVE is anyways necessary to assess hypertrophy of the FLR. So re-staging of the liver occurs max. 1-2 weeks before hepatic resection at our institution. Patients who showed unresectable progressive disease on these staging examinations (progressive disease as defined by RECIST criteria) would be no longer considered for surgery and receive alternative treatment options (usually chemotherapy). We have expanded the paragraph on “Post-interventional follow-up” (4.4., lines 239-241) in the Materials and Methods section to clarify this point.

Please add, if available, KRAS NRAS and BRAF status. Otherwise please add this as a strong limit of the authors’ findings interpretation.

Response:

We agree with you that tumor mutation status is a very important point. Unfortunately, since we are a tertiary care center and receive a lot of referrals specifically and solely for the purpose of surgical treatment of liver metastases, the histopathological work-up and reports are often performed outside of our institution and therefore not always documented in our electronic medical records.

In the patient cohort of this present study, KRAS status is available in 41/59 patients, and 12 out of these 41 had a positive KRAS mutation.

For right-sided colon cancer 6 patients had a KRAS mutation, 4 patient had no KRAS mutation and mutation status was unknown in the remaining 4 patients. For left-sided colon cancer, 6 patients had a KRAS mutation, 25 patient had no KRAS mutation and mutation status was unknown in the remaining 14 patients. So it appears that a KRAS mutation occurred more often in patients with right-sided colon cancer, which is consistent with previous studies 1,2 and could maybe be a reason for the worse outcome in patients with a right-sided cancer vs. left-sided cancer. We added this info in lines 70-77.

Info on N-RAS and B-RAF status as well as microsatellite instability is unfortunately even more limited for this patient cohort and we have added this as a major limitation to the appropriate paragraph at the end of the discussion section. (lines 195-197)

1 Xie MZ, Li JL, Cai ZM, Li KZ, Hu BL. Impact of primary colorectal Cancer location on the KRAS status and its prognostic value. BMC Gastroenterol. 2019;19(1):46. Published 2019 Mar 27. doi:10.1186/s12876-019-0965-5

2 Bylsma LC, Gillezeau C, Garawin TA, et al. Prevalence of RAS and BRAF mutations in metastatic colorectal cancer patients by tumor sidedness: A systematic review and meta-analysis. Cancer Med. 2020;9(3):1044‐1057. doi:10.1002/cam4.2747

Please add the types of chemotherapy performed.

Response:

The types of chemotherapy that were administered are mentioned in lines 90-91. Due to changing recommendations and individual tolerance/comorbidities/mutation status of patients, the types of chemotherapy used in this patient cohort varied quite a bit.

Which is the meaning of considering intrahepatic recurrence so important when the overall survival is not impacted by any of the variables considered?

Response:

We consider intrahepatic recurrence as important, since uncontrolled metastatic disease in the liver may threaten the function of the liver which in turn may significantly impact quality of life and survival (as opposed to lymph node metastases for example). However, we agree with you that overall survival is ultimately the most important outcome measure.

We found that Kaplan-Meier estimates for overall survival did also differ for patients with right-sided versus left-sided primary cancers (23.5 ± 11.6 versus 34.1 ± 3.5), although this did not reach statistical significance. Statistical significance was likely not reached due to insufficient number of endpoints: Only 25 of the 59 patients (42%) died, the rest is either still alive or lost to follow up. The number of known deaths is unfortunately particularly low in the group with right-sided colon cancer, leading to a high standard deviation of the estimated survival time in this subgroup. In summary however, we think that the statistically significant difference in intrahepatic recurrence and the tendency towards a significant difference for OS combined are strong evidence that tumor laterality may indeed have a prognostic value, possibly even for overall survival (although this needs to be confirmed with more data as mentioned in the paragraph on limitations at the end of the discussion section).

Please add the type of postoperative treatments. Were those patients who developed intrahepatic recurrence treated with more aggressive therapies? Otherwise, it seems that some others variables really impacted overall survival and their misidentification represents a strong bias – also for intrahepatic recurrence.

Response:

The decision whether patients would receive postoperative/adjuvant chemotherapy was based on the individual recommendation of a multidisciplinary tumor board for each patient. Since some patients therefore received adjuvant chemotherapy and others did not, we included this as an independent variable in the Cox regression analyses.

Further therapies after recurrence of disease included re-resection, local ablation, transarterial chemotherapy, transarterial radioembolization and palliative chemotherapy, depending on the extent and location of disease recurrence (we added this info in paragraph 4.5. lines 250-252). Of course, this means that the patient cohort is quite heterogeneous with regard to the postoperative/post-recurrence treatment, but in a retrospective study based on real-life data, this seems to be an unavoidable flaw. Nevertheless this obviously creates a significant bias for overall survival, hence why we acknowledged this as a significant limitation in the discussion section (lines 198-200).

It is unclear when bowel surgery was performed. Before? Before PVE but together with liver surgery? After liver surgery? This information is not secondary since the main authors’ finding regards the primary colic tumor.

Response:

In the majority of patients (47/59, 80%), resection of the primary tumor was performed before PVE and subsequent hepatic resection. In the remaining 12 patients (20%), a “liver-first approach” was pursued, and these patients were planned to have bowel surgery after PVE and right hemihepatectomy. We have added a paragraph to the results section (2.1.1.lines 78-83) containing this info.

Was adjuvant chemotherapy that therapy performed after the bowel surgery or after the liver surgery?

Response:

The terms “neoadjuvant” and “adjuvant” in the section on chemotherapy treatment refer to the temporal relation to liver surgery. We have edited the wording in this paragraph (2.1.3. lines 84-89) to make this less ambiguous.

Thank you very much for your time and effort to review our manuscript.

Round 2

Reviewer 1 Report

No further comments from this reviewer

Reviewer 2 Report

I have read again this revised manuscript, which has been substantially improved following reviewer's suggestions. I think that it will be of interest to the readers of Cancers.